# The Efficacy of FOLFIRI Plus Ramucirumab in Recurrent Colorectal Cancer Refractory to Adjuvant Chemotherapy with Oxaliplatin/Fluoropyrimidine—Including Biomarker Analyses

**DOI:** 10.3390/cancers17010091

**Published:** 2024-12-30

**Authors:** Naotoshi Sugimoto, Shingo Noura, Takeshi Kato, Shinichi Yoshioka, Taishi Hata, Atsushi Naito, Mitsuyoshi Tei, Hiroshi Tamagawa, Takamichi Komori, Yoshihito Ide, Takayuki Fukuzaki, Katsuki Danno, Genta Sawada, Yoshinori Kagawa, Toshio Shimokawa, Norikatsu Miyoshi, Takayuki Ogino, Mamoru Uemura, Hirofumi Yamamoto, Kohei Murata, Yuichiro Doki, Hidetoshi Eguchi

**Affiliations:** 1Department of Genetic Oncology, Osaka International Cancer Institute, Osaka 5418567, Japan; 2Department of Surgery, Sakai City Medical Center, Sakai 5938304, Japan; snoura0410@gmail.com; 3Department of Surgery, National Hospital Organization Osaka National Hospital, Osaka 5400006, Japan; kato.takeshi.hj@mail.hosp.go.jp; 4Department of Surgery, Yao Municipal Hospital, Yao 5810069, Japan; room335@mars.dti.ne.jp; 5Department of Surgery, Japan Organization of Occupational Health and Safety, Kansai Rosai Hospital, Amagasaki 6608511, Japan; hata-taishi@kansaih.johas.go.jp (T.H.); kmuratajp@yahoo.co.jp (K.M.); 6Department of Surgery, Osaka Police Hospital, Osaka 5438922, Japan; naito.atsu@gmail.com; 7Department of Surgery, Osaka Rosai Hospital, Sakai 5918025, Japan; mtei@osakah.johas.go.jp; 8Department of Gastrointestinal Surgery, Otemae Hospital, Osaka 5400008, Japan; htamagawa@otemae.gr.jp; 9Department of Surgery, Hyogo Prefectural Nishinomiya Hospital, Nishinomiya 6620918, Japan; t-komori@rd5.so-net.ne.jp; 10Department of Surgery, Japan Community Healthcare Organization Osaka Hospital, Osaka 5530003, Japan; ide-yoshihito@osaka.jcho.go.jp; 11Department of Gastroenterological Surgery, Osaka Saiseikai Senri Hospital, Suita 5650862, Japan; tafukuzaki@senri.saiseikai.or.jp; 12Department of Surgery, Minoh City Hospital, Minoh 5620014, Japan; danno_ka@yahoo.co.jp; 13Department of Surgery, Itami City Hospital, Itami 6648540, Japan; bluemayple2000@yahoo.co.jp; 14Department of Gastroenterological Surgery, Osaka General Medical Center, Osaka 5588558, Japan; yoshinori.kagawa@oici.jp; 15Clinical Study Support Center, Wakayama Medical University Hospital, Wakayama 6418509, Japan; toshibow2000@gmail.com; 16Department of Gastroenterological Surgery, Graduate School of Medicine, Osaka University, Suita 5650871, Japan; nmiyoshi@gesurg.med.osaka-u.ac.jp (N.M.); togino04@gesurg.med.osaka-u.ac.jp (T.O.); muemura@gesurg.med.osaka-u.ac.jp (M.U.); kobunyam@gmail.com (H.Y.); ydoki@gesurg.med.osaka-u.ac.jp (Y.D.); heguchi@gesurg.med.osaka-u.ac.jp (H.E.)

**Keywords:** FOLFIRI plus ramucirumab, no prior anti-angiogenic treatment, VEGF-D, TSP-2

## Abstract

There is no evidence on the efficacy of FOLFIRI plus ramucirumab in colorectal cancer refractory to oxaliplatin plus fluoropyrimidine without bevacizumab. No predictive marker has been established for ramucirumab, either. We planned a prospective study of the efficacy and toxicity of FOLFIRI plus ramucirumab refractory to adjuvant oxaliplatin plus fluoropyrimidine, adding an exploratory analysis to estimate the association of biomarkers with ramucirumab’s efficacy. A total of 48 patients were enrolled between September 2017 and September 2020. The median progression-free survival was 8.9 months (90% CI: 6.3–11.8), so the primary endpoint was met. The median overall survival and objective response rate were 22.3 months (95% CI: 17.4-NA) and 41.7% (95% CI: 4.9–7.6), respectively. The incidence of adverse events was consistent with that in previous reports. FOLFIRI plus ramucirumab is one of the options for recurrent colorectal cancer refractory to adjuvant oxaliplatin plus fluoropyrimidine. Serum VEGF-D levels may not be a predictive biomarker for ramucirumab. Serum TSP-2 may be a potential prognostic biomarker.

## 1. Introduction

The incidence of colorectal cancer is increasing every year in Japan. In 2019, colorectal cancer (CRC) was the most frequent cancer type and the second most common cause of cancer-related death in Japan [1]. In the EU and USA, CRC is the second most frequent cancer type, and the development of prevention, early diagnosis, and treatment methods for CRC is a very serious and indispensable issue.

The clinical usefulness of adjuvant chemotherapy for colon cancer has been reported in the EU and USA since the 1990s. The INT0035 study showed that the addition of fluorouracil (5-FU) and Levamisole (LEV) chemotherapy was superior to only surgery in terms of recurrence-free survival (RFS) and overall survival (OS) [2,3]. In addition, the results of the NT0089 study, the QUASAR study, and the NSABP C04 study were shown, and 5-FU/leucovorin (5FU/LV) therapy became a standard therapy as adjuvant chemotherapy for colon cancer [4,5,6,7]. Furthermore, as a result of the MOSAIC trial, which was carried out by de Gramont et al., FOLFOX (5-FU + leucovorin + oxaliplatin) therapy was deemed superior to 5FU/LV therapy in terms of disease-free survival (DFS) [8]. Integrating another two studies, the combination of 5-FU-type drugs and oxaliplatin has been recognized as a standard therapy [9,10]. The 3-year and 5-year DFS with FOLFOX is reported to be 78.2% and 73.3%, respectively. Sargent et al. undertook a meta-analysis of six large clinical trials in which 5-FU± irinotecan or oxaliplatin was administered and elucidated that recurrence was frequently observed within 3 years after the completion of adjuvant chemotherapy; specifically, the recurrence within 1 year was approximately 10% [11]. It is suggested that the selection of chemotherapy for these recurrent cases is important.

In the first-line setting for metastatic colorectal cancer patients, fluoropyrimidine, oxaliplatin, and bevacizumab combination therapy is one of the standard therapies. When a patient is refractory to fluoropyrimidine, oxaliplatin, and bevacizumab, FOLFIRI (5-FU + leucovorin + irinotecan) plus angiogenesis inhibitors is regarded the standard regimen. These angiogenesis inhibitors include bevacizumab, ziv-aflibercept, and ramucirumab. For recurrent cases, within 12 months after the completion of FOLFOX or CapeOX (capecitabine + oxaliplatin), FOLFIRI ± bevacizumab, ziv-aflibercept, or ramucirumab or FOLFIRI plus cetuximab or Panitumumab (*RAS* wild-type) is recommended in the NCCN guidelines [12]. The addition of bevacizumab or ziv-aflibercept to the standard cytotoxic drugs improved the survival time for patients refractory to first-line chemotherapy without bevacizumab [13,14]. Ramucirumab is a fully human immunoglobulin G1 monoclonal antibody VEGF (vascular endothelial growth factor) receptor-2 antagonist that prevents ligand binding and receptor-mediated pathway activation in endothelial cells. The RAISE study demonstrated that FOLFIRI plus ramucirumab significantly improved OS compared with FOLFIRI as a second-line treatment for patients with metastatic colorectal carcinoma refractory to fluoropyrimidine, OHP, and bevacizumab [15]. Moreover, a biomarker program identified VEGF-D as a potential predictive biomarker for ramucirumab’s efficacy in second-line mCRC prior to bevacizumab [16]. But in the RAISE study, every patient was refractory to fluoropyrimidine, oxaliplatin, and bevacizumab, and no study has prospectively evaluated FOLFIRI plus ramucirumab in patients refractory to fluoropyrimidine and oxaliplatin without bevacizumab for colorectal cancer.

Therefore, we planned a prospective study of the efficacy and toxicity of FOLFIRI plus ramucirumab in those refractory to adjuvant chemotherapy with oxaliplatin plus fluoropyrimidine without bevacizumab. Also, we designed a study of prospective biomarkers, estimating the association of biomarkers with ramucirumab’s efficacy.

## 2. Materials and Methods

### 2.1. Study Design

#### 2.1.1. The Main Study

The RAINCLOUD study was a multicenter, single-arm, phase II trial.

The eligibility criteria were as follows: (1) histologically or cytologically confirmed colorectal cancer; (2) confirmed recurrent colorectal cancer; (3) a history of receiving oxaliplatin and fluoropyrimidine as adjuvant chemotherapy and correspondence to either (a) experiencing radiographic recurrence during chemotherapy, (b) experiencing radiographic recurrence within 12 months after the discontinuation of chemotherapy due to adverse events, (c) experiencing radiographic recurrence within 12 months after the completion of scheduled chemotherapy, or (d) a doctor judging the patient to be intolerant to oxaliplatin; (4) having measurable disease based on the Response Evaluation Criteria in Solid Tumors, Version 1.1 (RECIST v 1.1); (5) being expected to survive more than 3 months; (6) having resolution to a grade ≤1, per the National Cancer Institute Common Terminology Criteria for Adverse Events, Version 4.03 (NCI-CTCAE v. 4.03), of all clinically significant toxic effects of prior chemotherapy, surgery, radiotherapy, or hormonal therapy, with the exception of peripheral neuropathy, which must have resolved to a grade ≤ 2; (7) an ECOG performance status of 0 or 1; (8) adequate hematologic function, as defined by an absolute neutrophil count (ANC) ≥ 1500/mm^3^, hemoglobin ≥ 9 g/dL, and platelets ≥ 100,000/mm^3^; (9) adequate coagulation function; (10) being clinically stable, asymptomatic, and adequately treated with anticoagulants; (11) adequate hepatic function, as defined by total bilirubin ≤ 1.5 mg/dL and aspartate aminotransferase (AST) and alanine aminotransferase (ALT) ≤ 100 IU/L (≤150 IU/L in the case of liver metastases); (12) adequate renal function, as defined by creatinine clearance > 50 mL/min; (13) urinary protein being less than 1+ on a dipstick or in a routine urinalysis; (14) having been notified about their disease and being able to provide signed informed consent; (15) being 20 years or older; and (16) agreeing to use adequate contraception methods (hormonal or barrier methods) during the study period and at least 12 weeks after the last dose of the study treatment or longer if required per the local regulations.

The exclusion criteria were as follows: (1) simultaneous or metachronous (disease-free duration within 5 years) double cancers except for intramucosal tumors or carcinoma curable in situ with local therapy; (2) a history of uncontrolled hereditary or acquired bleeding or thrombotic disorders; (3) an uncontrolled intercurrent illness, including, but not limited to, uncontrolled hypertension, symptomatic congestive heart failure (CHF), unstable angina pectoris, symptomatic or poorly controlled cardiac arrhythmia, inflammatory bowel disease, cirrhosis (Child B or worse), psychiatric illness/social disorders, or any other serious uncontrolled medical disorders in the opinion of the investigator; (4) Child B grade cirrhosis or worse, including any grade with a history of encephalopathy or clinically meaningful ascites resulting from cirrhosis (requiring diuretics or paracentesis); (5) experiencing any arterial thrombotic or arterial thromboembolic events; (6) receiving chronic antiplatelet therapy, including aspirin and nonsteroidal anti-inflammatory drugs; (7) known leptomeningeal disease or brain metastases or uncontrolled spinal cord compression; (8) an ongoing or active infection requiring parenteral antibiotic, antifungal, or antiviral therapy; (9) either HIV-1-antibody-, HIV-2-antibody-, or HBs-antigen- positive; (10) HBs-antibody- or HBc-antibody-positive and HBV-DNA-positive; (11) having received a prior autologous or allogeneic organ transplantation; (12) having undergone major surgery within the past 28 days; (13) having had a serious nonhealing wound, ulcer, or bone fracture within the past 28 days; (14) an elective or planned major surgery being set to be performed during the course of the trial; (15) having an acute or subacute bowel obstruction or a history of chronic diarrhea which is considered clinically significant in the opinion of the investigator; (16) having experienced a grade 3 or higher bleeding event within the past 3 months; (17) having either peptic ulcer disease associated with a bleeding event or known active diverticulitis; (18) having a known history or clinical evidence of Gilbert’s Syndrome or being known to have any of the genotypes UGT1A1*6/*6, UGT1A1*6/*28, or UGT1A1*28/*28; (19) having the desire to have children; and (20) participation in the trial being deemed inappropriate by a doctor.

FOLFIRI plus ramucirumab therapy was performed as follows: each 2-week cycle, the patients received 8 mg/kg ramucirumab as an intravenous infusion, followed by the FOLFIRI regimen (150 or 180 mg/m^2^ intravenous irinotecan concurrent with 200 mg/m^2^ intravenous leucovorin followed by 400 mg/m^2^ fluorouracil given as an intravenous bolus and then 2400 mg/m^2^ given as a continuous infusion over 48 h). The primary endpoint of this study was progression-free survival (PFS). Secondary endpoints were overall survival (OS), time to treatment failure (TTF), overall response rate (RR), disease control rate (DCR), and safety.

#### 2.1.2. Biomarker Study

Biomarker analyses were carried out to assess the correlations of the baseline individual marker levels with the clinical outcomes. Plasma samples were collected from whole blood before day 1 of cycle 1 (pre) and at the end of this study post discontinuation of treatment (post). We analyzed angiogenesis factors (HGF, PlGF, VEGF-A, VEGF-D, IFN-g, IL-6, IL-8, Angiopoietin-2, Neuropillin-1, Thrombospondin-2, OPN, sVEGFR1, sVEGFR2, sVEGFR3 sICAM-1, sVCAM-1, and TIMP-1) using a multiplex assay with Luminex™ technology (G&G Science Co.Ltd., Fukushima 9601242, Japan.).

### 2.2. Statistical Analysis

The primary endpoint was PFS. The FIRIS study showed that the PFS with FOLFIRI therapy of patients treated prior with oxaliplatin (patients treated prior with bevacizumab were not included) was 3.9 months [17]. The sub-analysis of the VELOUR study showed that the PFS of cases previously treated with a non-bevacizumab-based regimen was 6.9 months with FOLFIRI plus aflibercept [14]. Using the historical data as a comparison, when a threshold for the PFS in the protocol treatment (FOLFIRI plus ramucirumab) of 3.9 months and an expected PFS of 6.9 months were set and the one-sided alternative hypothesis H1, “PFS of this protocol treatment is more than 3.9 months”, with a significance level α = 0.1, was tested based on a statistical test of the population rate against the null hypothesis H0, “PFS of this protocol treatment is 3.9 months”, the minimum number of patients required to be registered was 44 in order for the power 1-β to outperform 0.8. Taking a few ineligible patients and dropout patients into consideration, the number of target patients was set to be 48.

The estimated curve for the PFS in the efficacy analysis set was calculated using the Kaplan–Meier estimate, and the 90% confidential interval was calculated using Greenwood’s formulation. The survival curve was graphed, and based on this, the median PFS and accumulated survival rate and yearly survival rate were estimated. The estimated curves for the OS and TTF in the efficacy analysis set were calculated using the Kaplan–Meier estimate, and the 95% confidential intervals were calculated using Greenwood’s formulation. The survival curve was graphed, and based on this, the median OS, TTF, and accumulated survival rate and yearly survival rate were estimated. The estimation of the response rate was based on RECIST 1.1. The 95% confidential interval for RR was calculated using the Clopper–Pearson exact test on the efficacy analysis set. The frequency of the best overall response was also calculated. The safety evaluation analysis was based on CTCAE v4.03. In the biomarker analyses, the cut-off values for dividing the patients into two groups for each marker were set as the medians.

This study was conducted in the MCSGO (Multicenter Clinical Study Group of Osaka) in Japan.

## 3. Results

### 3.1. Main Study Analysis

A total of 48 patients were enrolled from 15 sites between September 2017 and September 2020 (Figure 1). The major recruitment sites were as follows: Osaka University, Osaka International Cancer Institute, Sakai City Medical Center, National Hospital Organization Osaka National Hospital, Yao Municipal Hospital, at 11, 9, 7, 3, and 3. The patients’ characteristics were as follows: Median age: 63.5 years (25~77); male/female: 25/23; ECOG PS0/1: 44/4, right/left-sidedness: 10/38; stage at initial diagnosis, II/IIIA/IIIB/IV: 5/9/26/8; duration of adjuvant chemotherapy, <3M/3~6M: 8/40; duration of recurrence, < 12 months/>12 months@ 44/4; and *RAS* wild-type/mutant-type/unknown: 13/33/2 (Table 1). The median duration of the follow-up was 22.5 months. All 48 patients were included in the full set analysis. The relative dose intensities of the 5-FU bolus, the 5-FU continuous infusion, irinotecan, and ramucirumab were 74.1%, 88.3%, 87.7%, and 94.0%. The median PFS was 8.9 months (90% CI: 6.3–11.8), so the primary endpoint was met. The PFS rates at 6 months and 12 months were 66.7% (95% CI: 54.6–81.4) and 35.4% (95% CI: 24.2–51.9), respectively. The median OS was 22.3 months (95% CI: 17.4-NA) (Figure 2a,b), and the median TTF was 6.3 months (95% CI: 4.9–7.6). A total of 75% of the patients had been treated with at least one post-discontinuation anticancer therapy (PDT). The ORR and DCR were 41.7% and 81.3%, respectively.

An incidence of grade 3/4 adverse events of over 5% was seen for neutropenia (N = 20, 44%), leucopenia (N = 5, 10%), and hypertension (N = 4, 8%). No unexpected adverse events or treatment-related deaths were observed (Table 2).

### 3.2. Biomarker Analysis

In total, 81 plasma samples were collected from 48 patients. The numbers of samples at the time points of pre-/post- treatment were 45/36, and 32 patients had samples both pre- and post-treatment.

Post-treatment, the levels of VEGF-A, VEGF-D, PlGF, and TSP-2 were all significantly higher than those pre-treatment (Figure 3). For the other parameters, post-treatment, the levels of sVCAM-1 and sVEGFR3 rose and fell compared to those pre-treatment.

Next, to verify the relationships of the parameters with the efficacy outcomes, we defined the high and low groups as being above and below the median plasma levels.

In the biomarker analysis populations, in terms of the response rate, the low-PlGF, low-HGF, and low-TSP-2 groups had better response rates than the equivalent groups with high levels (60.0% vs. 28.6%, odds ratio: 3.62 (0.86–16.96); 63.2% vs. 27.3%, odds ratio: 4.39 (1.03–21.19); and 66.7% vs. 20.0%, odds ratio: 7.54 (1.62–43.79)), but for the other biomarkers, including VEGF-A and VEGF-D, there were no differences between the low and high groups.

In both the PFS and OS analyses, in terms of VEGF-D, there were no differences between the low and high groups (10.4 months vs. 6.5 months, HR: 0.86 (0.46–1.61); 24.3 months vs. 22.4 months, HR: 0.82 (0.39–1.73)).

In terms of VEGF-A, there were no differences between the low and high groups either (10.4 months vs. 8.1 months, HR: 0.79 (0.41–1.51); 26.0 months vs. 21.8 months, HR: 0.73 (0.34–1.59)).

In terms of PlGF, the low group had a longer PFS than the high group (16.0 months vs. 6.3 months, HR: 0.47 (0.25-0.90)), but there was no difference in OS (25.2 months vs. 22.1 months, HR: 0.82 (0.39-1.73)).

Only for TSP-2 levels did the low group have a survival benefit over the high group (14.5 months vs. 6.0 months, HR: 0.40; 28.1 months vs. 16.6 months, HR: 0.49) (Figure 4 and Figure 5).

For the other biomarkers, including sVCAM-1 and sVEGFR3, there were no differences between the low and high groups in terms of PFS and OS.

## 4. Discussion

This is the first prospective report on the efficacy of FOLFIRI plus ramucirumab in patients with cancer refractory to oxaliplatin plus fluoropyrimidine in the adjuvant setting.

The present results demonstrated the median PFS was 8.9 months, the median OS was 22.3 months, and the ORR was 41.7% for patients refractory to oxaliplatin plus fluoropyrimidine without prior treatment with bevacizumab. A total of 79% of the patients had only one metastatic site and 75% of the patients had been treated with PDT in our study. In the RAISE study, the PFS, OS, and ORR were 5.7 months, 13.3 months, and 13.4% in the FOLFIRI plus ramucirumab arm in patients refractory to oxaliplatin, fluoropyrimidine, and bevacizumab. A total of 32% of the patients had only one metastatic site and 57.1% of the patients had been treated with PDT in the RAISE study. For cases treated with bevacizumab and ziv-aflibercept, the data on its efficacy in patients refractory to first-line chemotherapy were better without bevacizumab than with bevacizumab [13,14]. Two characteristics of the patients’ backgrounds—most of the patients had only one metastatic site and all of the patients were treated prior without bevacizumab—and a high percentage of PDT may have affected the good therapeutic effect in our study.

In the NCCN guidelines, for patients with recurrent metastatic diseases previously treated with FOLFOX or CapeOx within the past 12 months, FOLFIRI plus anti-angiogenic agents is recommended. But there are no prospective data on this. For bevacizumab, there were no data on its use in combination with FOLFIRI. For ziv-aflibercept, only a subset analysis existed. Our study provides the first prospective data on the efficacy of FOLFIRI plus anti-angiogenesis treatments without prior bevacizumab treatment. So, FOLFIRI plus ramucirumab is the one of the options for cancer refractory to oxaliplatin plus fluoropyrimidine in the adjuvant setting.

The adverse events observed in our study were consistent with those in the RAISE study. All grades of hematological toxicities were found more frequently than in the RAISE study, especially leucopenia and anemia, but the rate of grade 3–4 events was not so high. Moreover, febrile neutropenia was not observed. In terms of non-hematological toxicities, the rate of grade 3–4 toxicities was low, including diarrhea and fatigue. All adverse events of special interest with ramucirumab were reported at an incidence lower than 10%. So, FOLFIRI plus ramucirumab was tolerable in this setting.

In our biomarker study, serum VEGF-D seemed not to be a predictive biomarker, but TSP-2 may be a potential prognostic biomarker for ramucirumab. In the RAISE biomarker study, higher levels of VEGF-D expression were deemed a potential predictive biomarker for ramucirumab’s efficacy [16]. But according to recent reports using a multiplex assay, there were no differences in the PFS and OS between patients with higher levels of VEGF-D and PlGF and those with lower levels [18,19]. Also, in our study, there were no differences in the PFS and OS between those with high and low VEGF-D. So, VEGF-D is regarded as a prognostic biomarker but not one that is predictive of ramucirumab’s efficacy.

In terms of PlGF, the PFS in the low group was longer than that in the high group, but the OS was not longer. As we have described, recent reports show there is no difference in the PFS and OS between patients with high and low PlGF. We cannot deny that the results for PlGF are down to chance due to the small number of studies.

In terms of TSP-2, both the PFS and OS in the low group were longer than those in the high group.

Thrombospondins (THBS, TSP) are a family of five secreted matricellular glycoproteins that broadly regulate cell–matrix interactions, angiogenesis, cell proliferation, and apoptosis [20,21,22,23]. Among all of the thrombospondins, TSP-2 has been most commonly studied in cancer diagnosis and progression. Circulating levels of TSP-2 were confirmed to be a potential diagnostic candidate in pancreatic cancer and lung cancer [24,25,26,27], while the histological expression of TSP-2 has been identified to be an independent prognostic biomarker for distal cholangiocarcinoma, colorectal cancer, and urothelial carcinoma [28,29,30]. Some other studies have also noted that TSP-2 may act as a useful salivary marker for the detection of oral cavity squamous cell carcinoma [31]. However, few studies have evaluated the predictive and prognostic value of THBS2 in triple-negative breast cancer or in a neoadjuvant setting specifically. Gao reviewed that patients with increased TSP-2 had a shorter OS (HR = 1:64; 95% CI = 1:21–2.22); however, no difference was found in the DFS between groups with high and low TSP-2 (HR = 1:44; 95% CI = 0:28–7.33) [32].

As previously reported, patients with high TSP-2 saw a worse efficacy of FOLFIRI plus ramucirumab in terms of their RR, PFS, and OS in our study, so TSP-2 may still be a prognostic biomarker for metastatic colorectal cancer.

This study has some limitations. Firstly, this was a single-arm, phase II study, so the results were exploratory. The number of patients was small, and the possibility of selection bias cannot be denied. Secondly, some of the patients had unknown BRAF and MSI statuses because these analyses had not been approved in Japan when this study launched in 2017. Finally, a U-KIT analysis is yet to be explored, and the cut-off lines for the biomarkers are not still validated.

Also, there were no prospective data for CRC patients refractory to adjuvant chemotherapy with oxaliplatin plus fluoropyrimidine. Moreover, there were no data for an analysis of the relationship between VEGF-D expression and efficacy in patients with colorectal cancers who had not been treated prior with anti-angiogenic treatments.

## 5. Conclusions

FOLFIRI plus ramucirumab is one of the treatment options for patients with recurrent colorectal cancer refractory to oxaliplatin plus fluoropyrimidine without bevacizumab.

Serum VEGF-D levels may not be a predictive biomarker for ramucirumab’s efficacy. Serum TSP-2 may be a potential prognostic biomarker. Further studies in larger cohorts are needed to confirm our results.

## Figures and Tables

**Figure 1 cancers-17-00091-f001:**
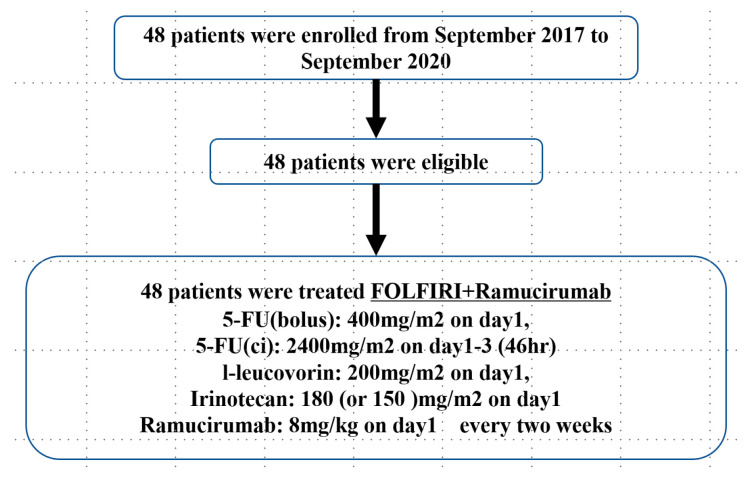
CONSORT flowchart showing the disposition of the enrolled patients at the time of the data design.

**Figure 2 cancers-17-00091-f002:**
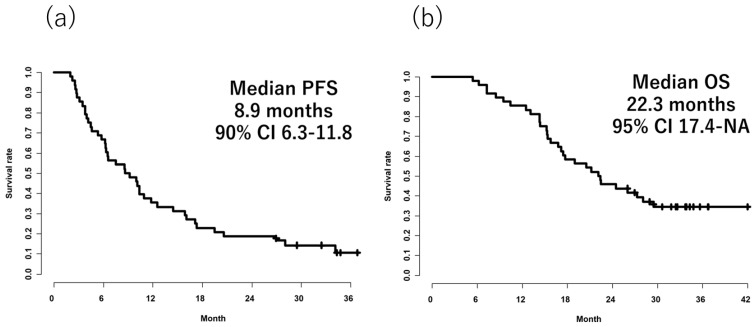
Kaplan–Meier survival analysis of (**a**) PFS, progression-free survival, and (**b**) OS, overall survival; CI, confidence interval.

**Figure 3 cancers-17-00091-f003:**
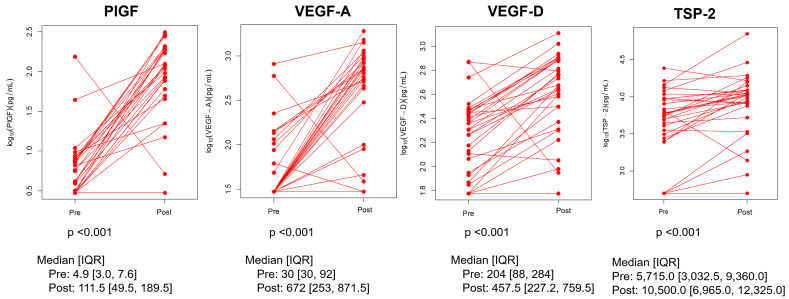
Dynamics of angiogenesis factors. PlGF: placental growth factor; VEGF-A: vascular endothelial growth factor A; VEGF-D: vascular endothelial growth factor D; TSP-2: Thrombospondin-2.

**Figure 4 cancers-17-00091-f004:**
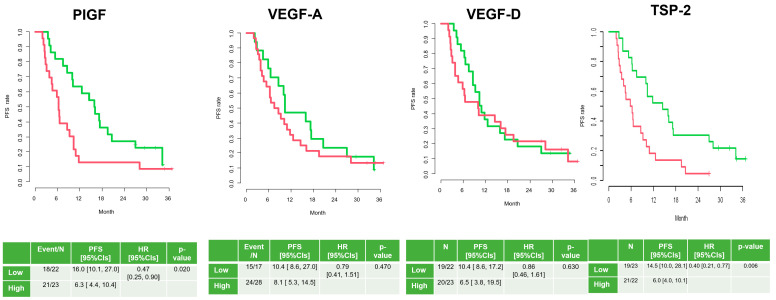
Progression-free survival according to angiogenesis factors pre-treatment (high = red; low = green).

**Figure 5 cancers-17-00091-f005:**
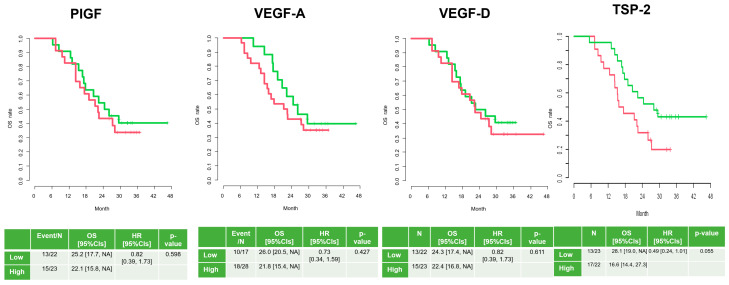
Overall survival according to angiogenesis factors pre-treatment (high = red; low = green).

**Table 1 cancers-17-00091-t001:** Baseline characteristics of patients.

		N = 48	
**Age (Years)**	Median (range)	63.5	(25~77)
**Gender**	Male	25	52%
	Female	23	48%
**ECOG PS**	0	44	92%
	1	4	8%
**Primary Lesion**	Right	10	21%
	Left	38	79%
**Histology**	Differentiated	42	88%
	Undifferentiated	6	12%
**Stage at Initial Diagnosis**	II	5	10%
	IIIA	9	19%
	IIIB	26	54%
	IV	8	17%
**Surgical Curability**	Cur A	38	79%
	Cur B	9	19%
	Cur C	1	2%
**Duration of Recurrence**	<12 months	44	92%
	>12 months	4	8%
**Oxaliplatin**	Refractory	44	92%
	Intolerant	4	8%
**Metastatic sites**	Lung	11	23%
	Liver	21	44%
	Lymph nodes	7	15%
	Peritoneum	16	33%
**Number of Metastatic Sites**	1	38	79%
	2	8	17%
	>3	2	4%
**Ras Status**	Wild-type	13	27%
	Mutant-type	33	69%
	Unknown	2	4%

**Table 2 cancers-17-00091-t002:** Treatment-related adverse events in safety analysis set.

	Adverse Events	All Grades		≥G3	
**Hematological**	Leucopenia	28	58%	5	10%
	Neutropenia	27	56%	20	44%
	Anemia	31	65%	1	2%
	Thrombocytopenia	18	38%	2	4%
**Non-Hematological**	Nausea	18	36%	1	2%
	Vomiting	8	17%	1	2%
	Anorexia	21	44%	1	2%
	Diarrhea	14	29%	1	2%
	Infection	1	2%	1	2%
	Pneumonitis	1	2%	0	0%
	Fatigue	22	46%	2	4%
	Mucositis	8	17%	2	4%
	Alopecia	11	23%	-	-
	Increased ALT	20	42%	2	4%
**Of Special Interest**	Hypertension	11	23%	4	8%
	Proteinuria	22	46%	1	2%
	Thromboembolic Events	1	2%	0	0%

ALT: alanine aminotransferase.

## Data Availability

The dataset is available on request from the authors. Clinical Trial Information: UMIN000028678.

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
