# Peer review of "The Efficacy of FOLFIRI Plus Ramucirumab in Recurrent Colorectal Cancer Refractory to Adjuvant Chemotherapy with Oxaliplatin/Fluoropyrimidine—Including Biomarker Analyses"

_cancers, 2024, doi:10.3390/cancers17010091_

Round 1

Reviewer 1 Report

Comments and Suggestions for Authors

Reviewer Report:

The authors have performed interesting studies on CRC and carefully delineated the identification of TSP-2 as a potential prognostic marker for recurrent colorectal cancer. A combination of  FOLFIRI  and ramucirumab is found to be one of the treatment options for patients with recurrent colorectal cancer that is refractory to adjuvant chemotherapy with oxaliplatin plus fluoropyrimidine. Probably the initial studies needs verification from a large cohort of patients in future.

The manuscript can be published in the journal with the following points that needs to taken care by the authors.

Language: The presentation of the results needs to be improved thoroughly. For example, the following lines, a few examples in the manuscript, needs to be modified without any doubt. However the manuscript should be thoroughly polished for presentation in proper English.

Line no. 77-78, "In anti-VEGF...." sentence is broken and incomplete

Line no. 124-125 ""showed that FOLFORI plus... treatment"

Line no. 132-133 Change "ineligibility" to "ineligible"

Line no. 212-214. Short and broken sentences.

Line no. 244-245 be precise and clear

Line no. 260-261 to be rewritten

Line no. 266 Replace "secondary" with secondly

Comments on the Quality of English Language

Please see the comments given in the reviewers report. The presentation language of the manuscript needs thorough improvement.

Reviewer 2 Report

Comments and Suggestions for Authors

Authors planned prospective study of the efficacy and toxicity of FOLFIRI plus ramucirumab refractory to adjuvant chemotherapy with oxaliplatin plus fluoropyrimidine. Also, they designed as a prospective biomarker study estimating the association of biomarkers with ramucirumab efficacy. This manuscript can be accepted after revision.

Here are the points:

1.      There should be more detail about the patients (such as if they have other cronic disorders, or if they use other drugs, which affect the suggested treatment).

2.      Introduction is very brief. It should be expanded.

3.      “caluculated” should be written as “calculated”.

4.      They indicated that a total of 48 patients were enrolled from 15 sites. Authors specifically mentioned of which sites.

Comments on the Quality of English Language

Authors planned prospective study of the efficacy and toxicity of FOLFIRI plus ramucirumab refractory to adjuvant chemotherapy with oxaliplatin plus fluoropyrimidine. Also, they designed as a prospective biomarker study estimating the association of biomarkers with ramucirumab efficacy. This manuscript can be accepted after revision.

Here are the points:

1.      There should be more detail about the patients (such as if they have other cronic disorders, or if they use other drugs, which affect the suggested treatment).

2.      Introduction is very brief. It should be expanded.

3.      “caluculated” should be written as “calculated”.

4.      They indicated that a total of 48 patients were enrolled from 15 sites. Authors specifically mentioned of which sites.

Reviewer 3 Report

Comments and Suggestions for Authors

 The authors investigated efficacy and toxicity of FOLFIRI plus Ramucirumab in patients with recurrent colorectal cancer after adjuvant chemotherapy. They conducted a multicenter prospective study in patients with recurrent colorectal cancer who were resistant to adjuvant chemotherapy with oxaliplatin and fluoropyrimidine. The clinical trial was a single-arm phase II study. The primary endpoint was progression-free survival (PFS), and the secondary ones were overall survival (OS), time to treatment failure (TTF), overall response rate (RR), and disease control rate (DCR). In addition, serum biomarkers were examined before and after treatment. In the results, this study showed median OS of 22.3 months and median TTF of 6.3 months, with acceptable toxicity of treatment. Biomarker analysis showed that patients with high pre-treatment PIGF and TSP had better FPS, and patients with high TSP levels had better OS. They conducted a multicenter prospective study to explore effective treatments for a patient population for which there is currently no standard treatment. However, this paper has several major flaws.

1. The objective of this study, evaluation of the efficacy and toxicity of FOLFIRI plus Ramucirumab treatment, has already been confirmed in other clinical trials with different target patient groups, and there is little point in designing the study as a single-arm phase II trial.

2. The introduction is not organized. Instead of dividing paragraphs into small sections, they should organize the points you want to make in each paragraph.

3. Although the standard treatment is described, there is no mention at all of how treatment is currently being carried out for patients who have relapsed after adjuvant treatment, which is the subject of their study, and what problems exist. They should clarify the target of their research and provide detailed background.

4. They failed to adequately describe the patient population included in their study in Material and Method section.

5. They claim that the study targets treatment-resistant cases after adjuvant chemotherapy, but there is no standard for how they determine treatment resistance. In addition, it should be noted that these are cases in which surgery was performed on the primary tumor.

6. Case inclusion and exclusion criteria should be clearly stated.

7. In Figure 1, only eligible cases are listed; excluded cases should also be listed appropriately.

8. In the patient background information in Table 1, the stage of disease at initial diagnosis, duration of adjuvant chemotherapy, and time to recurrence must also be clearly indicated.

9. The criteria for determining whether the biomarkers listed in the results are high or low should also be clearly stated in the Method.

10. In the discussion, comparisons of OS, etc. with clinical trials with different patient backgrounds should be refrained from as they may mislead readers.

11. Abbreviations should be properly spelled out, e.g., pts.

12. Treatment regimens should also be listed when first written, e.g., FOLFIRI (5-FU+l-LV+CPT-11).

Comments on the Quality of English Language

The authors had better improve the overall structure of the text rather than individual sentences.

Round 2

Reviewer 3 Report

Comments and Suggestions for Authors

The authors successfully revised the manuscript in accordance with all comments.